# STOCHASTIC LATENT ACTOR-CRITIC: DEEP REINFORCEMENT LEARNING WITH A LATENT VARIABLE MODEL

## ABSTRACT

Deep reinforcement learning (RL) algorithms can use high-capacity deep networks to learn directly from image observations. However, these kinds of observation spaces present a number of challenges in practice, since the policy must now solve two problems: a representation learning problem, and a task learning problem. In this paper, we aim to explicitly learn representations that can accelerate reinforcement learning from images. We propose the stochastic latent actor-critic (SLAC) algorithm: a sample-efficient and high-performing RL algorithm for learning policies for complex continuous control tasks directly from high-dimensional image inputs. SLAC learns a compact latent representation space using a stochastic sequential latent variable model, and then learns a critic model within this latent space. By learning a critic within a compact state space, SLAC can learn much more efficiently than standard RL methods. The proposed model improves performance substantially over alternative representations as well, such as variational autoencoders. In fact, our experimental evaluation demonstrates that the sample efficiency of our resulting method is comparable to that of model-based RL methods that directly use a similar type of model for control. Furthermore, our method outperforms both model-free and model-based alternatives in terms of final performance and sample efficiency, on a range of difficult image-based control tasks. Our code and videos of our results are available at our website.[1]

## 1 INTRODUCTION

Deep reinforcement learning (RL) algorithms can automatically learn to solve certain tasks from raw, low-level observations such as images. However, these kinds of observation spaces present a number of challenges in practice: on one hand, it is difficult to directly learn from these high-dimensional inputs, but on the other hand, it is also difficult to tease out a compact representation of the underlying task-relevant information from which to learn instead. For these reasons, deep RL directly from low-level observations such as images remains a challenging problem. Particularly in continuous domains governed by complex dynamics, such as robotic control (Tassa et al., 2018; Brockman et al., 2016), standard approaches still require separate sensor setups to monitor details of interest in the environment, such as the joint positions of a robot or specific pose information of objects of interest. To instead be able to learn directly from the more general and rich modality of vision would greatly advance the current state of our learning systems, so we aim to study precisely this. Standard model-free deep RL aims to use direct end-to-end training to explicitly unify these tasks of representation learning and task learning. However, solving both problems together is difficult, since an effective policy requires an effective representation, but in order for an effective representation to emerge, the policy or value function must provide meaningful gradient information using only the model-free supervision signal (i.e., the reward function). In practice, learning directly from images with standard RL algorithms can be slow, sensitive to hyperparameters, and inefficient. In contrast to end-to-end learning with RL, predictive learning can benefit from a rich and informative supervision signal before the agent has even made progress on the task or received any rewards. This leads us to ask: can we explicitly *learn* a latent representation from raw low-level observations that makes deep RL easier, through learning a predictive latent variable model?

---

[1] https://rl-slac.github.io/slac/

Predictive models are commonly used in model-based RL for the purpose of planning (Deisenroth & Rasmussen, 2011; Finn & Levine, 2017; Nagabandi et al., 2018; Chua et al., 2018; Zhang et al., 2019) or generating cheap synthetic experience for RL to reduce the required amount of interaction with the real environment (Sutton, 1991; Gu et al., 2016). However, in this work, we are primarily concerned with their potential to alleviate the *representation learning* challenge in RL. We devise a stochastic predictive model by modeling the high-dimensional observations as the consequence of a latent process, with a Gaussian prior and latent dynamics, as illustrated in Figure 1. A model with an entirely stochastic latent state has the appealing interpretation of being able to properly represent uncertainty about any of the state variables, given its past observations. We demonstrate in our work that fully stochastic state space models can in fact be learned effectively: With a well-designed stochastic network, such models outperform fully deterministic models, and contrary to the observations in prior work (Hafner et al., 2019; Buesing et al., 2018), are actually comparable to partially stochastic models. Finally, we note that this explicit representation learning, even on low-reward data, allows an agent with such a model to make progress on representation learning even before it makes progress on task learning.

Equipped with this model, we can then perform RL *in the learned latent space* of the predictive model. We posit—and confirm experimentally—that our latent variable model provides a useful representation for RL. Our model represents a partially observed Markov decision process (POMDP), and solving such a POMDP exactly would be computationally intractable (Astrom, 1965; Kaelbling et al., 1998; Igl et al., 2018). We instead propose a simple approximation that trains a Markovian critic on the (stochastic) latent state and trains an actor on a history of observations and actions. The resulting stochastic latent actor-critic (SLAC) algorithm loses some of the benefits of full POMDP solvers, but it is easy and stable to train. It also produces good results, in practice, on a range of challenging problems, making it an appealing alternative to more complex POMDP solution methods.

The main contributions of our SLAC algorithm are useful representations learned from our stochastic sequential latent variable model, as well as effective RL in this learned latent space. We show experimentally that our approach substantially improves on *both* model-free and model-based RL algorithms on a range of image-based continuous control benchmark tasks, attaining better final performance and learning more quickly than algorithms based on (a) end-to-end deep RL from images, (b) learning in a latent space produced by various alternative latent variable models, such as a variational autoencoder (VAE) (Kingma & Welling, 2014), and (c) model-based RL based on latent state-space models with partially stochastic variables (Hafner et al., 2019).

## 2 RELATED WORK

**Representation learning in RL.** End-to-end deep RL can in principle learn representations directly as part of the RL process (Mnih et al., 2013). However, prior work has observed that RL has a "representation learning bottleneck": a considerable portion of the learning period must be spent acquiring good representations of the observation space (Shelhamer et al., 2016). This motivates the use of a distinct representation learning procedure to acquire these representations before the agent has even learned to solve the task. The use of auxiliary supervision in RL to learn such representations has been explored in a number of prior works (Lange & Riedmiller, 2010; Finn et al., 2016; Jaderberg et al., 2017; Higgins et al., 2017; Ha & Schmidhuber, 2018; Nair et al., 2018; Oord et al., 2018; Gelada et al., 2019; Dadashi et al., 2019). In contrast to this class of representation learning algorithms, we explicitly learn a latent variable model of the POMDP, in which the latent representation and latent-space dynamics are jointly learned. By modeling covariances between consecutive latent states, we make it feasible for our proposed algorithm to perform Bellman backups directly in the latent space of the learned model.

**Partial observability in RL.** Our work is also related to prior research on RL under partial observability. Prior work has studied exact and approximate solutions to POMDPs, but they require explicit models of the POMDP and are only practical for simpler domains (Kaelbling et al., 1998). Recent work has proposed end-to-end RL methods that use recurrent neural networks to process histories of observations and (sometimes) actions, but without constructing a model of the POMDP (Hausknecht & Stone, 2015; Foerster et al., 2016; Zhu et al., 2018). Other works, however, learn latent-space dynamical system models and then use them to solve the POMDP with model-based RL (Watter et al., 2015; Wahlström et al., 2015; Karl et al., 2017; Zhang et al., 2019; Hafner et al., 2019).

Although some of these works learn latent variable models that are similar to ours, these model-based methods are often limited by compounding model errors and finite horizon optimization. In contrast to these works, our approach does not use the model for prediction and performs infinite horizon policy optimization. Our approach benefits from the good asymptotic performance of model-free RL, while at the same time leveraging the improved latent space representation for sample efficiency. Other works have also trained latent variable models and used their representations as the inputs to model-free RL algorithms. They use representations encoded from latent states sampled from the forward model (Buesing et al., 2018), belief representations obtained from particle filtering (Igl et al., 2018), or belief representations obtained directly from a learned belief-space forward model (Gregor et al., 2019). Our approach is closely related to these prior methods, in that we also use model-free RL with a latent state representation that is learned via prediction. However, instead of using belief representations, our method learns a critic directly on latent states samples.

**Sequential latent variable models.** Several previous works have explored various modeling choices to learn stochastic sequential models (Krishnan et al., 2015; Archer et al., 2015; Karl et al., 2016; Fraccaro et al., 2016; 2017; Doerr et al., 2018a). In the context of using sequential models for RL, previous works have typically observed that partially stochastic state space models are more effective than fully stochastic ones (Buesing et al., 2018; Igl et al., 2018; Hafner et al., 2019). In these models, the state of the underlying MDP is modeled with the deterministic state of a recurrent network (e.g., LSTM (Hochreiter & Schmidhuber, 1997) or GRU (Cho et al., 2014)), and optionally with some stochastic random variables. As mentioned earlier, a model with a latent state that is entirely stochastic has the appealing interpretation of learning a representation that can properly represent uncertainty about any of the state variables, given past observations. We demonstrate in our work that fully stochastic state space models can in fact be learned effectively and, with a well-designed stochastic network, such models perform on par to partially stochastic models and outperform fully deterministic models.

# 3 REINFORCEMENT LEARNING AND MODELING

This work addresses the problem of learning maximum entropy policies from high-dimensional observations in POMDPs, by simultaneously learning a latent representation of the underlying MDP state using variational inference and learning the policy in a maximum entropy RL framework. In this section, we describe maximum entropy RL (Ziebart, 2010; Haarnoja et al., 2018a; Levine, 2018) in fully observable MDPs, as well as variational methods for training latent state space models for POMDPs.

## 3.1 MAXIMUM ENTROPY RL IN FULLY OBSERVABLE MDPs

In a Markov decision process (MDP), an agent at time $t$ takes an action $\mathbf{a}_t \in \mathcal{A}$ from state $\mathbf{s}_t \in \mathcal{S}$ and reaches the next state $\mathbf{s}_{t+1} \in \mathcal{S}$ according to some stochastic transition dynamics $p(\mathbf{s}_{t+1}|\mathbf{s}_t, \mathbf{a}_t)$. The initial state $\mathbf{s}_1$ comes from a distribution $p(\mathbf{s}_1)$, and the agent receives a reward $r_t$ on each of the transitions. Standard RL aims to learn the parameters $\phi$ of some policy $\pi_\phi(\mathbf{a}_t|\mathbf{s}_t)$ such that the expected sum of rewards is maximized under the induced trajectory distribution $\rho_\pi$. This objective can be modified to incorporate an *entropy* term, such that the policy also aims to maximize the expected entropy $\mathcal{H}(\pi_\phi(\cdot|\mathbf{s}_t))$ under the induced trajectory distribution $\rho_\pi$. This formulation has a close connection to variational inference (Ziebart, 2010; Haarnoja et al., 2018a; Levine, 2018), and we build on this in our work. The resulting maximum entropy objective is

$$\phi^* = \arg\max_\phi \sum_{t=1}^{T} \mathbb{E}_{(\mathbf{s}_t, \mathbf{a}_t) \sim \rho_\pi} [r(\mathbf{s}_t, \mathbf{a}_t) + \alpha \mathcal{H}(\pi_\phi(\cdot|\mathbf{s}_t))], \quad (1)$$

where $r$ is the reward function, and $\alpha$ is a temperature parameter that controls the trade-off between optimizing for the reward and for the entropy (i.e., stochasticity) of the policy. Soft actor-critic (SAC) (Haarnoja et al., 2018a) uses this maximum entropy RL framework to derive soft policy iteration, which alternates between policy evaluation and policy improvement within the described maximum entropy framework. SAC then extends this soft policy iteration to handle continuous action spaces by using parameterized function approximators to represent both the Q-function $Q_\theta$ (critic) and the policy $\pi_\phi$ (actor). The soft Q-function parameters $\theta$ are optimized to minimize the

soft Bellman residual,

$$J_Q(\theta) = \mathbb{E}_{(\mathbf{s}_t, \mathbf{a}_t, r_t, \mathbf{s}_{t+1}) \sim \mathcal{D}} \left[ \frac{1}{2} \left( Q_\theta(\mathbf{s}_t, \mathbf{a}_t) - (r_t + \gamma V_{\bar{\theta}}(\mathbf{s}_{t+1})) \right)^2 \right], \quad (2)$$

$$V_{\bar{\theta}}(\mathbf{s}_{t+1}) = \mathbb{E}_{\mathbf{a}_{t+1} \sim \pi_\phi} \left[ Q_{\bar{\theta}}(\mathbf{s}_{t+1}, \mathbf{a}_{t+1}) - \alpha \log \pi_\phi(\mathbf{a}_{t+1} | \mathbf{s}_{t+1}) \right], \quad (3)$$

where $\mathcal{D}$ is the replay buffer, $\gamma$ is the discount factor, and $\bar{\theta}$ are delayed parameters. The policy parameters $\phi$ are optimized to update the policy towards the exponential of the soft Q-function,

$$J_\pi(\phi) = \mathbb{E}_{\mathbf{s}_t \sim \mathcal{D}} \left[ \mathbb{E}_{\mathbf{a}_t \sim \pi_\phi} \left[ \alpha \log(\pi_\phi(\mathbf{a}_t | \mathbf{s}_t)) - Q_\theta(\mathbf{s}_t, \mathbf{a}_t) \right] \right]. \quad (4)$$

Results of this stochastic, entropy maximizing RL framework demonstrate improved robustness and stability. SAC also shows the sample efficiency benefits of an off-policy learning algorithm, in conjunction with the high performance benefits of a long-horizon planning algorithm. Precisely for these reasons, we choose to extend the SAC algorithm in this work to formulate our SLAC algorithm.

### 3.2 SEQUENTIAL LATENT VARIABLE MODELS AND AMORTIZED VARIATIONAL INFERENCE IN POMDPS

To learn representations for RL, we use latent variable models trained with amortized variational inference. The learned model must be able to process a large number of pixels that are present in the entangled image $\mathbf{x}$, and it must tease out the relevant information into a compact and disentangled representation $\mathbf{z}$. To learn such a model, we can consider maximizing the probability of each observed datapoint $\mathbf{x}$ from some training set $\mathcal{D}$ under the entire generative process $p(\mathbf{x}) = \int p(\mathbf{x}|\mathbf{z})p(\mathbf{z}) \, d\mathbf{z}$. This objective is intractable to compute in general due to the marginalization of the latent variables $\mathbf{z}$. In amortized variational inference, we utilize the following bound on the log-likelihood (Kingma & Welling, 2014),

$$E_{\mathbf{x} \sim D} \left[ \log p(\mathbf{x}) \right] \geq E_{\mathbf{x} \sim D} \left[ E_{\mathbf{z} \sim q} \left[ \log p(\mathbf{x}|\mathbf{z}) \right] - \mathrm{D}_{\mathrm{KL}} \left( q(\mathbf{z}|\mathbf{x}) \parallel p(\mathbf{z}) \right) \right]. \quad (5)$$

We can maximize the probability of the observed datapoints (i.e., the left hand side of Equation (5)) by learning an encoder $q(\mathbf{z}|\mathbf{x})$ and a decoder $p(\mathbf{x}|\mathbf{z})$, and then directly performing gradient ascent on the right hand side of the equation. In this setup, the distributions of interest are the prior $p(\mathbf{z})$, the observation model $p(\mathbf{x}|\mathbf{z})$, and the posterior $q(\mathbf{z}|\mathbf{x})$.

Although such generative models have been shown to successfully model various types of complex distributions (Kingma & Welling, 2014) by embedding knowledge of the distribution into an informative latent space, they do not have a built-in mechanism for the use of temporal information when performing inference. In the case of partially observable environments, as we discuss below, the representative latent state $\mathbf{z}_t$ corresponding to a given non-Markovian observation $\mathbf{x}_t$ needs to be informed by past observations.

Consider a partially observable MDP (POMDP), where an action $\mathbf{a}_t \in \mathcal{A}$ from latent state $\mathbf{z}_t \in \mathcal{Z}$ results in latent state $\mathbf{z}_{t+1} \in \mathcal{Z}$ and emits a corresponding observation $\mathbf{x}_{t+1} \in \mathcal{X}$. We make an explicit distinction between an observation $\mathbf{x}_t$ and the underlying latent state $\mathbf{z}_t$, to emphasize that the latter is unobserved and the distribution is not known a priori. Analogous to the fully observable MDP, the initial state distribution is $p(\mathbf{z}_1)$, the transition probability distribution is $p(\mathbf{z}_{t+1}|\mathbf{z}_t, \mathbf{a}_t)$, and the reward is $r_t$. In addition, the observation model is given by $p(\mathbf{x}_t|\mathbf{z}_t)$.

As in the case for VAEs, a generative model of these observations $\mathbf{x}_t$ can be learned by maximizing the log-likelihood. In the POMDP setting, however, we note that $\mathbf{x}_t$ alone does not provide all necessary information to infer $\mathbf{z}_t$, and thus, prior temporal information must be taken into account. This brings us to the discussion of sequential latent variable models. The distributions of interest are the priors $p(\mathbf{z}_1)$ and $p(\mathbf{z}_{t+1}|\mathbf{z}_t, \mathbf{a}_t)$, the observation model $p(\mathbf{x}_t|\mathbf{z}_t)$, and the approximate posteriors $q(\mathbf{z}_1|\mathbf{x}_1)$ and $q(\mathbf{z}_{t+1}|\mathbf{x}_{t+1}, \mathbf{z}_t, \mathbf{a}_t)$. The log-likehood of the observations can then be bounded, similarly to the VAE bound in Equation (5), as

$$\log p(\mathbf{x}_{1:\tau+1}|\mathbf{a}_{1:\tau}) \geq \mathbb{E}_{\mathbf{z}_{1:\tau+1} \sim q} \left[ \sum_{t=1}^{\tau+1} \log p(\mathbf{x}_t|\mathbf{z}_t) - \mathrm{D}_{\mathrm{KL}} \left( q(\mathbf{z}_1|\mathbf{x}_1) \parallel p(\mathbf{z}_1) \right) \right.$$

$$\left. - \sum_{t=1}^{\tau} \mathrm{D}_{\mathrm{KL}} \left( q(\mathbf{z}_{t+1}|\mathbf{x}_{t+1}, \mathbf{z}_t, \mathbf{a}_t) \parallel p(\mathbf{z}_{t+1}|\mathbf{z}_t, \mathbf{a}_t) \right) \right]. \quad (6)$$

Prior work (Hafner et al., 2019; Buesing et al., 2018; Doerr et al., 2018b) has explored modeling such non-Markovian observation sequences, using methods such as recurrent neural networks with deterministic hidden state, as well as probabilistic state-space models. In this work, we enable the effective training of a fully stochastic sequential latent variable model, and bring it together with a maximum entropy actor-critic RL algorithm to create SLAC: a sample-efficient and high-performing RL algorithm for learning policies for complex continuous control tasks directly from high-dimensional image inputs.

## 4 JOINT MODELING AND CONTROL AS INFERENCE

Our method aims to learn maximum entropy policies from high-dimensional, non-Markovian observations in a POMDP, while also learning a model of that POMDP. The model alleviates the representation learning problem, which in turn helps with the policy learning problem. We formulate the control problem as inference in a probabilistic graphical model with latent variables, as shown in Figure 1.

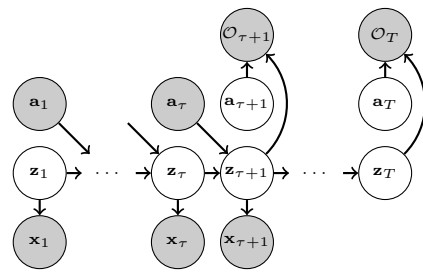

Figure 1: Graphical model of POMDP with optimality variables for $t \geq \tau + 1$.

For a fully observable MDP, the control problem can be embedded into a graphical model by introducing a binary random variable $\mathcal{O}_t$, which indicates if time step $t$ is optimal. When its distribution is chosen to be $p(\mathcal{O}_t = 1 | \mathbf{s}_t, \mathbf{a}_t) = \exp(r(\mathbf{s}_t, \mathbf{a}_t))$, then maximization of $p(\mathcal{O}_{1:T})$ via approximate inference in that model yields the optimal policy for the maximum entropy objective (Levine, 2018).

In a POMDP setting, the distribution can analogously be given by $p(\mathcal{O}_t = 1 | \mathbf{z}_t, \mathbf{a}_t) = \exp(r(\mathbf{z}_t, \mathbf{a}_t))$. Instead of maximizing the likelihood of the optimality variables alone, we jointly model the observations (including the observed rewards of the past time steps) and learn maximum entropy policies by maximizing the marginal likelihood $p(\mathbf{x}_{1:\tau+1}, \mathcal{O}_{\tau+1:T} | \mathbf{a}_{1:\tau})$. This objective represents both the likelihood of the observed data from the past $\tau$ steps, as well as the optimality of the agent's actions for future steps. We factorize our variational distribution into a product of *recognition* terms $q(\mathbf{z}_1 | \mathbf{x}_1)$ and $q(\mathbf{z}_{t+1} | \mathbf{x}_{t+1}, \mathbf{z}_t, \mathbf{a}_t)$, *dynamics* terms $p(\mathbf{z}_{t+1} | \mathbf{z}_t, \mathbf{a}_t)$, and *policy* terms $\pi(\mathbf{a}_t | \mathbf{z}_t)$:

$$q(\mathbf{z}_{1:T}, \mathbf{a}_{\tau+1:T} | \mathbf{x}_{1:\tau+1}, \mathbf{a}_{1:\tau})$$

$$= q(\mathbf{z}_1 | \mathbf{x}_1) \prod_{t=1}^{\tau} q(\mathbf{z}_{t+1} | \mathbf{x}_{t+1}, \mathbf{z}_t, \mathbf{a}_t) \prod_{t=\tau+1}^{T-1} p(\mathbf{z}_{t+1} | \mathbf{z}_t, \mathbf{a}_t) \prod_{t=\tau+1}^{T} \pi(\mathbf{a}_t | \mathbf{z}_t). \quad (7)$$

The variational distribution uses the dynamics for future time steps to prevent the agent from controlling the transitions and from choosing optimistic actions (Levine, 2018). The posterior over the actions represents the agent's policy $\pi$. Although this derivation uses a policy that is conditioned on the latent state, our algorithm, which will be described in the next section, learns a parametric policy that is directly conditioned on observations and actions. This approximation allows us to directly execute the policy without having to perform inference on the latent state at run time.

We use the posterior from Equation (7) to obtain the evidence lower bound (ELBO) of the marginal likelihood,

$$\log p(\mathbf{x}_{1:\tau+1}, \mathcal{O}_{\tau+1:T} | \mathbf{a}_{1:\tau}) \geq \mathbb{E}_{(\mathbf{z}_{1:T}, \mathbf{a}_{\tau+1:T}) \sim q} \left[ \sum_{t=1}^{\tau+1} \log p(\mathbf{x}_t | \mathbf{z}_t) \right.$$

$$- \mathrm{D}_{\mathrm{KL}} \left( q(\mathbf{z}_1 | \mathbf{x}_1) \, \| \, p(\mathbf{z}_1) \right) - \sum_{t=1}^{\tau} \mathrm{D}_{\mathrm{KL}} \left( q(\mathbf{z}_{t+1} | \mathbf{x}_{t+1}, \mathbf{z}_t, \mathbf{a}_t) \, \| \, p(\mathbf{z}_{t+1} | \mathbf{z}_t, \mathbf{a}_t) \right)$$

$$\left. + \sum_{t=\tau+1}^{T} \left( r(\mathbf{z}_t, \mathbf{a}_t) + \log p(\mathbf{a}_t) - \log \pi(\mathbf{a}_t | \mathbf{z}_t) \right) \right], \quad (8)$$

where $p(\mathbf{a}_t)$ is the action prior. The full derivation of the ELBO is given in Appendix A. This derivation assumes that the reward function, which determines $p(\mathcal{O}_t | \mathbf{z}_t, \mathbf{a}_t)$, is known. However, in many RL problems, this is not the case. In that situation, we can simply append the reward to the

observation, and learn the reward along with $p(\mathbf{x}_t|\mathbf{z}_t)$. This requires no modification to our method other than changing the observation space, and we use this approach in all of our experiments. We do this to learn latent representations that are more relevant to the task, but we do not use predictions from it. Instead, the RL objective uses rewards from the agent's experience, as in model-free RL.

## 5 STOCHASTIC LATENT ACTOR CRITIC

We now describe our stochastic latent actor critic (SLAC) algorithm, which approximately maximizes the ELBO using function approximators to model the prior and posterior distributions. The ELBO objective in Equation (8) can be split into a model objective and a maximum entropy RL objective. The model objective can directly be optimized, while the maximum entropy RL objective can be solved via message passing. We can learn Q-functions for the messages, and then we can rewrite the RL objective to express it in terms of these messages. Additional details of the derivation of the SLAC objectives are given in Appendix A.

**Latent Variable Model**: The first part of the ELBO corresponds to training the latent variable model to maximize the likelihood of the observations, analogous to the ELBO in Equation (6) for the sequential latent variable model. The distributions of the latent variable model are diagonal Gaussian distributions, where the means and variances are outputs of neural networks. The distribution parameters $\psi$ of this model are optimized to maximize the first part of the ELBO. The model loss is

$$J_M(\psi) = \mathbb{E}_{(\mathbf{x}_{1:\tau+1}, \mathbf{a}_{1:\tau}, r_{1:\tau}) \sim \mathcal{D}} \left[ \mathbb{E}_{\mathbf{z}_{1:\tau+1} \sim q_\psi} \left[ \sum_{t=1}^{\tau+1} \log p_\psi(\mathbf{x}_t|\mathbf{z}_t) \right. \right.$$
$$\left. \left. - D_{\mathrm{KL}}\left(q_\psi(\mathbf{z}_1|\mathbf{x}_1) \,\|\, p_\psi(\mathbf{z}_1)\right) - \sum_{t=1}^{\tau} D_{\mathrm{KL}}\left(q_\psi(\mathbf{z}_{t+1}|\mathbf{x}_{t+1}, \mathbf{z}_t, \mathbf{a}_t) \,\|\, p_\psi(\mathbf{z}_{t+1}|\mathbf{z}_t, \mathbf{a}_t)\right) \right] \right]. \quad (9)$$

We use the reparameterization trick to sample from the filtering distribution $q_\psi(\mathbf{z}_{1:\tau+1}|\mathbf{x}_{1:\tau+1}, \mathbf{a}_{1:\tau})$.

**Critic and Actor**: The second part of the ELBO corresponds to the maximum entropy RL objective. As in the fully observable case from Section 3.1 and as described by Levine (2018), this optimization can be solved via message passing of soft Q-values, except that we use the latent states $\mathbf{z}$ rather than the true states $\mathbf{s}$. For continuous state and action spaces, this message passing is approximated by minimizing the soft Bellman residual, which we use to train our soft Q-function parameters $\theta$,

$$J_Q(\theta) = \mathbb{E}_{(\mathbf{x}_{1:\tau+1}, \mathbf{a}_{1:\tau}, r_\tau) \sim \mathcal{D}} \left[ \mathbb{E}_{\mathbf{z}_{1:\tau+1} \sim q_\psi} \left[ \frac{1}{2} \left( Q_\theta(\mathbf{z}_\tau, \mathbf{a}_\tau) \right. \right. \right.$$
$$\left. \left. \left. - \left( r_\tau + \gamma \mathbb{E}_{\mathbf{a}_{\tau+1} \sim \pi_\phi} [Q_{\bar{\theta}}(\mathbf{z}_{\tau+1}, \mathbf{a}_{\tau+1}) - \alpha \log \pi_\phi(\mathbf{a}_{\tau+1}|\mathbf{x}_{1:\tau+1}, \mathbf{a}_{1:\tau})] \right) \right)^2 \right] \right], \quad (10)$$

where $\bar{\theta}$ are delayed parameters, obtained as exponential moving averages of $\theta$. Notice that the latents $\mathbf{z}_\tau$ and $\mathbf{z}_{\tau+1}$, which are used in the Bellman backup, are sampled from the same joint, i.e. $\mathbf{z}_{\tau+1} \sim q_\psi(\mathbf{z}_{\tau+1}|\mathbf{x}_{\tau+1}, \mathbf{z}_\tau, \mathbf{a}_\tau)$. The RL objective, which corresponds to the second part of the ELBO, can be rewritten in terms of the soft Q-function. The policy parameters $\phi$ are optimized to maximize this objective, analogously to soft actor-critic (Haarnoja et al., 2018a). The policy loss is then

$$J_\pi(\phi) = \mathbb{E}_{(\mathbf{x}_{1:\tau+1}, \mathbf{a}_{1:\tau}) \sim \mathcal{D}} \left[ \mathbb{E}_{\mathbf{z}_{1:\tau+1} \sim q_\psi} \left[ \mathbb{E}_{\mathbf{a}_{\tau+1} \sim \pi_\phi} [\alpha \log \pi_\phi(\mathbf{a}_{\tau+1}|\mathbf{x}_{1:\tau+1}, \mathbf{a}_{1:\tau}) - Q_\theta(\mathbf{z}_{\tau+1}, \mathbf{a}_{\tau+1})] \right] \right]. \quad (11)$$

We assume a uniform action prior, so $p(\mathbf{a}_t)$ is a constant term that we omit from the policy loss. We use the reparameterization trick to sample from the policy, and the policy loss only uses the last sample $\mathbf{z}_{\tau+1}$ of the sequence for the critic. Although the policy used in our derivation is conditioned in the latent state, our learned *parametric* policy is conditioned directly on the past observations and actions, so that the learned policy can be executed at run time without requiring inference of the latent state. Finally, we note that for the expectation over latent states in the Bellman residual in Equation (10), rather than sampling latent states $\mathbf{z} \sim \mathcal{Z}$, we sample latent states from the filtering distribution $q_\psi(\mathbf{z}_{1:\tau+1}|\mathbf{x}_{1:\tau+1}, \mathbf{a}_{1:\tau})$. This design choice allows us to minimize the critic loss for samples that are most relevant for $Q$, while also allowing the critic loss to use the Q-function in the same way as implied by the policy loss in Equation (11).

SLAC is outlined in Algorithm 1. The actor-critic component follows prior work, with automatic tuning of the temperature $\alpha$ and two Q-functions to mitigate underestimation (Fujimoto et al., 2018; Haarnoja et al., 2018a;b). SLAC can be viewed as a variant of SAC (Haarnoja et al., 2018a) where the critic is trained on the stochastic latent state of our sequential latent variable model. The backup for the critic is performed on a tuple $(\mathbf{z}_\tau, \mathbf{a}_\tau, r_\tau, \mathbf{z}_{\tau+1})$, sampled from the posterior $q(\mathbf{z}_{\tau+1}, \mathbf{z}_\tau | \mathbf{x}_{1:\tau+1}, \mathbf{a}_{1:\tau})$. The critic can, in principle, take advantage of the perfect knowledge of the state $\mathbf{z}_t$, which makes learning easier. However, the parametric policy does not have access to $\mathbf{z}_t$, and must make decisions based on a history of observations and actions. SLAC is not a model-based algorithm, in that in does not use the model for prediction, but we see in our experiments that SLAC can achieve similar sample efficiency as a model-based algorithm.

---

**Algorithm 1** Stochastic Latent Actor-Critic (SLAC)

---

**Require:** $E, \psi, \theta_1, \theta_2, \phi$      ▷ Environment and initial parameters for model, actor, and critic
    $\mathbf{x}_1 \sim E_{\text{reset}}()$      ▷ Sample initial observation from the environment
    $\mathcal{D} \leftarrow (\mathbf{x}_1)$      ▷ Initialize replay buffer with initial observation
    **for** each iteration **do**
        **for** each environment step **do**
            $\mathbf{a}_t \sim \pi_\phi(\mathbf{a}_t | \mathbf{x}_{1:t}, \mathbf{a}_{1:t-1})$      ▷ Sample action from the policy
            $r_t, \mathbf{x}_{t+1} \sim E_{\text{step}}(\mathbf{a}_t)$      ▷ Sample transition from the environment
            $\mathcal{D} \leftarrow \mathcal{D} \cup (\mathbf{a}_t, r_t, \mathbf{x}_{t+1})$      ▷ Store the transition in the replay buffer
        **for** each gradient step **do**
            $\psi \leftarrow \psi - \lambda_M \nabla_\psi J_M(\psi)$      ▷ Update model weights
            $\theta_i \leftarrow \theta_i - \lambda_Q \nabla_{\theta_i} J_Q(\theta_i)$ for $i \in \{1, 2\}$      ▷ Update the Q-function weights
            $\phi \leftarrow \phi - \lambda_\pi \nabla_\phi J_\pi(\phi)$      ▷ Update policy weights
            $\bar{\theta}_i \leftarrow \nu \theta_i + (1 - \nu)\bar{\theta}_i$ for $i \in \{1, 2\}$      ▷ Update target critic network weights

---

## 6   Latent Variable Model

We briefly summarize our full model architecture here, with full details in Appendix B. Motivated by the recent success of autoregressive latent variables in VAEs (Razavi et al., 2019; Maaloe et al., 2019), we factorize the latent variable $\mathbf{z}_t$ into two stochastic layers, $\mathbf{z}_t^1$ and $\mathbf{z}_t^2$, as shown in Figure 2. This factorization results in latent distributions that are more expressive, and it allows for some parts of the prior and posterior distributions to be shared. We found this design to produce high quality reconstructions and samples, and utilize it in all of our experiments. The generative model $p$ and the inference model $q$ are given by

$$p_\psi(\mathbf{z}_1) = p_\psi(\mathbf{z}_1^2 | \mathbf{z}_1^1) p(\mathbf{z}_1^1),$$
$$p_\psi(\mathbf{z}_{t+1} | \mathbf{z}_t, \mathbf{a}_t) = p_\psi(\mathbf{z}_{t+1}^2 | \mathbf{z}_{t+1}^1, \mathbf{z}_t^2, \mathbf{a}_t) p_\psi(\mathbf{z}_{t+1}^1 | \mathbf{z}_t^2, \mathbf{a}_t),$$
$$q_\psi(\mathbf{z}_1 | \mathbf{x}_1) = p_\psi(\mathbf{z}_1^2 | \mathbf{z}_1^1) q_\psi(\mathbf{z}_1^1 | \mathbf{x}_1),$$
$$q_\psi(\mathbf{z}_{t+1} | \mathbf{x}_{t+1}, \mathbf{z}_t, \mathbf{a}_t) = p_\psi(\mathbf{z}_{t+1}^2 | \mathbf{z}_{t+1}^1, \mathbf{z}_t^2, \mathbf{a}_t) q_\psi(\mathbf{z}_{t+1}^1 | \mathbf{x}_{t+1}, \mathbf{z}_t^2, \mathbf{a}_t).$$

Note that we choose the variational distribution $q$ over $\mathbf{z}_t^2$ to be the same as the model $p$. Thus, the KL divergence in $J_M$ simplifies to the divergence between $q$ and $p$ over $\mathbf{z}_t^1$. We use a multivariate standard normal distribution for $p(\mathbf{z}_1^1)$, since it is not conditioned on any variables, i.e. $\mathbf{z}_1^1 \sim \mathcal{N}(\mathbf{0}, \mathbf{I})$. The conditional distributions of our model are diagonal Gaussian, with means and variances given by neural networks. Unlike models from prior work (Hafner et al., 2019; Buesing et al., 2018; Doerr et al., 2018b), which have deterministic and stochastic paths and use recurrent neural networks, ours is fully stochastic, i.e. our latent state is a Markovian latent random variable formed by the concatenation of $\mathbf{z}_t^1$ and $\mathbf{z}_t^2$. Further details are discussed in Appendix B.

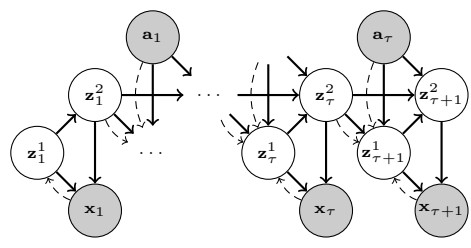

Figure 2: Diagram of our full model. Solid arrows show the generative model, dashed arrows show the inference model. Rewards are not shown for clarity.

## 7    EXPERIMENTAL EVALUATION

We evaluate SLAC on numerous image-based continuous control tasks from both the DeepMind Control Suite (Tassa et al., 2018) and OpenAI Gym (Brockman et al., 2016), as illustrated in Figure 3. Full details of SLAC's network architecture are described in Appendix B. Aside from the value of action repeats (i.e. control frequency) for the tasks, we kept all of SLAC's hyperparameters constant across all tasks in all domains. Training and evaluation details are given in Appendix C, and image samples from our model for all tasks are shown in Appendix D. Additionally, visualizations of our results and code are available on the project website.[2]

### 7.1    COMPARATIVE EVALUATION ON CONTINUOUS CONTROL BENCHMARK TASKS

To provide a comparative evaluation against prior methods, we evaluate SLAC on four tasks (cheetah run, walker walk, ball-in-cup catch, finger spin) from the DeepMind Control Suite (Tassa et al., 2018), and four tasks (cheetah, walker, ant, hopper) from OpenAI Gym (Brockman et al., 2016). Note that the Gym tasks are typically used with low-dimensional state observations, while we evaluate on them with raw image observations. We compare our method to the following state-of-the-art model-based and model-free algorithms:

**SAC** (Haarnoja et al., 2018a): This is an off-policy actor-critic algorithm, which represents a comparison to state-of-the-art model-free learning. We include experiments showing the performance of SAC based on true state (as an upper bound on performance) as well as directly from raw images.

**MPO** (Abdolmaleki et al., 2018b;a): This is an off-policy actor-critic algorithm that performs an expectation maximization form of policy iteration, learning directly from raw images.

**D4PG** (Barth-Maron et al., 2018): This is also an off-policy actor-critic algorithm, learning directly from raw images. The results reported in the plots below are the performance after $10^8$ training steps, as stated in the benchmarks from (Tassa et al., 2018).

**PlaNet** (Hafner et al., 2019): This is a model-based RL method for learning from images, which uses a partially stochastic sequential latent variable model, but without explicit policy learning. Instead, the model is used for planning with model predictive control (MPC), where each plan is optimized with the cross entropy method (CEM).

**DVRL** (Igl et al., 2018): This is an on-policy model-free RL algorithm that also trains a partially stochastic latent-variable POMDP model. DVRL uses the *full belief* over the latent state as input into both the actor and critic, as opposed to our method, which trains the critic with the latent state and the actor with a history of actions and observations.

Our experiments on the DeepMind Control Suite in Figure 4 show that the sample efficiency of SLAC is comparable or better than *both* model-based and model-free alternatives. This indicates that overcoming the representation learning bottleneck, coupled with efficient off-policy RL, provides for fast learning similar to model-based methods, while attaining final performance comparable to fully model-free techniques that learn from state. SLAC also substantially outperforms DVRL. This difference can be explained in part by the use of an efficient off-policy RL algorithm, which can better take advantage of the learned representation.

We also evaluate SLAC on continuous control benchmark tasks from OpenAI Gym in Figure 5. We notice that these tasks are much more challenging than the DeepMind Control Suite tasks, because the rewards are not as shaped and not bounded between 0 and 1, the dynamics are different, and the episodes terminate on failure (e.g., when the hopper or walker falls over). PlaNet is unable to solve

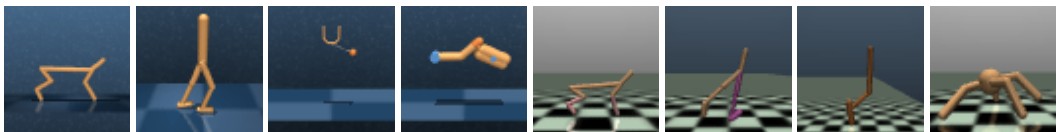

Figure 3: Example image observations for our continuous control benchmark tasks: DeepMind Control's cheetah run, walker walk, ball-in-cup catch, and finger spin, and OpenAI Gym's half cheetah, walker, hopper, and ant (left to right). Images are rendered at a resolution of $64 \times 64$ pixels.

---

[2]https://rl-slac.github.io/slac/

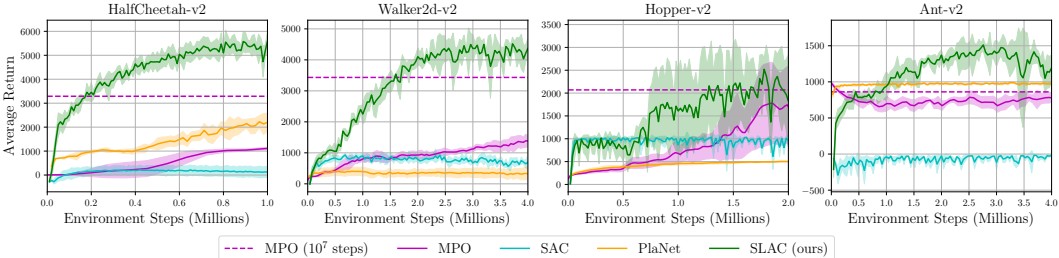

Figure 4: Experiments on the DeepMind Control Suite from images (unless otherwise labeled as "state"). SLAC (ours) converges to similar or better final performance than the other methods, while almost always achieving reward as high as the upper bound SAC baseline that learns from true state. Note that for these experiments, 1000 environments steps corresponds to 1 episode.

Figure 5: Experiments on the OpenAI Gym benchmark tasks from images. SLAC (ours) converges to higher performance than both PlaNet and SAC on all four of these tasks. The number of environments steps in each episode is variable, depending on the termination.

the last three tasks, while for the cheetah task, it learns a suboptimal policy that involves flipping the cheetah over and pushing forward while on its back. To better understand the performance of fixed-horizon MPC on these tasks, we also evaluated with the ground truth dynamics (i.e., the true simulator), and found that even in this case, MPC did not achieve good final performance, suggesting that infinite horizon policy optimization, of the sort performed by SLAC and model-free algorithms, is important to attain good results on these tasks.

Our experiments show that SLAC successfully learns complex continuous control benchmark tasks from raw image inputs. On the DeepMind Control Suite, SLAC exceeds the performance of PlaNet on three of the tasks, and matches its performance on the walker task. However, on the harder image-based OpenAI Gym tasks, SLAC outperforms PlaNet by a large margin. In both domains, SLAC substantially outperforms all prior model-free methods. We note that the prior methods that we tested generally performed poorly on the image-based OpenAI Gym tasks, despite considerable hyperparameter tuning.

## 7.2 EVALUATING THE LATENT VARIABLE MODEL

We next study the tradeoffs between different design choices for the latent variable model. We compare our **fully stochastic** model, as described in Section 6, to a standard non-sequential **VAE** model (Kingma & Welling, 2014), which has been used in multiple prior works for representation learning in RL (Higgins et al., 2017; Ha & Schmidhuber, 2018; Nair et al., 2018), the partially stochastic model used by **PlaNet** (Hafner et al., 2019), as well as three variants of our model: a **simple filtering** model that does not factorize the latent variable into two layers of stochastic units, a **fully deterministic** model that removes all stochasticity from the hidden state dynamics, and a **partially stochastic** model that has both deterministic and stochastic transitions, similar to the PlaNet model, but with our architecture. Both the fully deterministic and partially stochastic models use the same architecture as our fully stochastic model, including the same

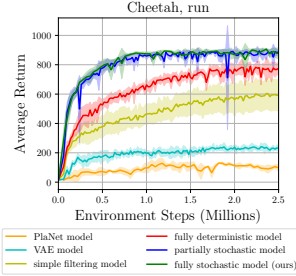

Figure 6: Comparison of different design choices for the latent variable model.

two-level factorization of the latent variable. In all cases, we use the RL framework of SLAC and only vary the choice of model for representation learning. As shown in the comparison in Figure 6, our fully stochastic model outperforms prior models as well as the deterministic and simple variants

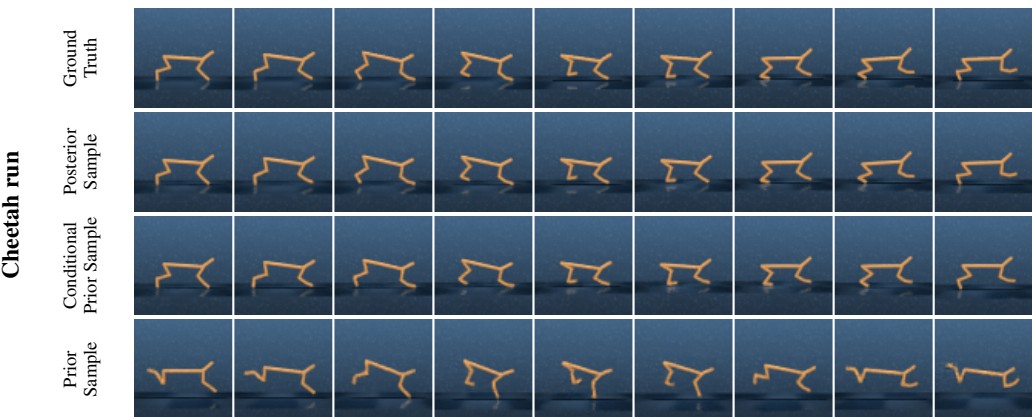

Figure 7: Example image sequence seen for the cheetah task (first row), corresponding posterior sample (reconstruction) from our model (second row), and generated prediction from the generative model (last two rows). The second to last row is conditioned on the first frame (i.e., the posterior model is used for the first time step while the prior model is used for all subsequent steps), whereas the last row is not conditioned on any ground truth images. Note that all of these sampled sequences are conditioned on the same action sequence, and that our model produces highly realistic samples, even when predicting via the generative model.

of our own model. The partially stochastic variant of our model matches the performance of our fully stochastic model but, contrary to the conclusions in prior work (Hafner et al., 2019; Buesing et al., 2018), the fully stochastic model performs on par, while retaining the appealing interpretation of a stochastic state space model. We hypothesize that these prior works benefit from the deterministic paths (realized as an LSTM or GRU) because they use multi-step samples from the prior. In contrast, our method uses samples from the posterior, which are conditioned on same-step observations, and thus these latent samples are less sensitive to the propagation of the latent states through time.

### 7.3 Qualitative Predictions from the Latent Variable Model

We show example image samples from our learned sequential latent variable model for the cheetah task in Figure 7, and we include the other tasks in Appendix D. Samples from the posterior show the images $\mathbf{x}_t$ as constructed by the decoder $p_\psi(\mathbf{x}_t|\mathbf{z}_t)$, using a sequence of latents $\mathbf{z}_t$ that are encoded and sampled from the posteriors, $q_\psi(\mathbf{z}_1|\mathbf{x}_1)$ and $q_\psi(\mathbf{z}_{t+1}|\mathbf{x}_{t+1}, \mathbf{z}_t, \mathbf{a}_t)$. Samples from the prior, on the other hand, use a sequence of latents where $\mathbf{z}_1$ is sampled from $p(\mathbf{z}_1)$ and all remaining latents $\mathbf{z}_t$ are from the propagation of the previous latent state through the latent dynamics $p_\psi(\mathbf{z}_{t+1}|\mathbf{z}_t, \mathbf{a}_t)$. Note that these prior samples do not use any image frames as inputs, and thus they do not correspond to any ground truth sequence. We also show samples from the conditional prior, which is conditioned on the first image from the true sequence: for this, the sampling procedure is the same as the prior, except that $\mathbf{z}_1$ is encoded and sampled from the posterior $q_\psi(\mathbf{z}_1|\mathbf{x}_1)$, rather than being sampled from $p(\mathbf{z}_1)$. We notice that the generated images samples can be sharper and more realistic by using a smaller variance for $p_\psi(\mathbf{x}_t|\mathbf{z}_t)$ when training the model, but at the expense of a representation that leads to lower returns. Finally, note that we do not actually use the samples from the prior for training.

## 8 Discussion

We presented SLAC, an efficient RL algorithm for learning from high-dimensional image inputs that combines efficient off-policy model-free RL with representation learning via a sequential stochastic state space model. Through representation learning in conjunction with effective task learning in the learned latent space, our method achieves improved sample efficiency and final task performance as compared to *both* prior model-based and model-free RL methods.

While our current SLAC algorithm is fully model-free, in that predictions from the model are not utilized to speed up training, a natural extension of our approach would be to use the model predictions themselves to generate synthetic samples. Incorporating this additional synthetic model-based data into a mixed model-based/model-free method could further improve sample efficiency and performance. More broadly, the use of explicit representation learning with RL has the potential to not only accelerate training time and increase the complexity of achievable tasks, but also enable reuse and transfer of our learned representation across tasks.

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

## A    DERIVATION OF THE EVIDENCE LOWER BOUND AND SLAC OBJECTIVES

In this appendix, we discuss how the SLAC objectives can be derived from applying a variational inference scheme to the control as inference framework for reinforcement learning (Levine, 2018) . In this framework, the problem of finding the optimal policy is cast as an inference problem, conditioned on the evidence that the agent is behaving optimally. While Levine (2018) derives this in the fully observed case, we present a derivation in the POMDP setting.

We aim to maximize the marginal likelihood $p(\mathbf{x}_{1:\tau+1}, \mathcal{O}_{\tau+1:T}|\mathbf{a}_{1:\tau})$, where $\tau$ is the number of steps that the agent has already taken. This likelihood reflects that the agent cannot modify the past $\tau$ actions and they might have not been optimal, but it can choose the future actions up to the end of the episode, such that the chosen future actions are optimal. Notice that unlike the standard control as inference framework, in this work we not only maximize the likelihood of the optimality variables but also the likelihood of the observations, which provides additional supervision for the latent representation. This does not come up in the MDP setting since the state representation is fixed and learning a dynamics model of the state would not change the model-free equations derived from the maximum entropy RL objective.

For reference, we restate the factorization of our variational distribution:

$$q(\mathbf{z}_{1:T}, \mathbf{a}_{\tau+1:T}|\mathbf{x}_{1:\tau+1}, \mathbf{a}_{1:\tau})$$
$$= q(\mathbf{z}_1|\mathbf{x}_1) \prod_{t=1}^{\tau} q(\mathbf{z}_{t+1}|\mathbf{x}_{t+1}, \mathbf{z}_t, \mathbf{a}_t) \prod_{t=\tau+1}^{T-1} p(\mathbf{z}_{t+1}|\mathbf{z}_t, \mathbf{a}_t) \prod_{t=\tau+1}^{T} \pi(\mathbf{a}_t|\mathbf{z}_t). \quad (12)$$

As discussed by Levine (2018), the agent does not have control over the stochastic dynamics, so we use the dynamics $p(\mathbf{z}_{t+1}|\mathbf{z}_t, \mathbf{a}_t)$ for $t \geq \tau + 1$ in the variational distribution in order to prevent the agent from choosing optimistic actions.

The joint likelihood is

$$p(\mathbf{x}_{1:\tau+1}, \mathcal{O}_{\tau+1:T}, \mathbf{z}_{1:T}, \mathbf{a}_{\tau+1:T}|\mathbf{a}_{1:\tau})$$
$$= p(\mathbf{z}_1) \prod_{t=1}^{T-1} p(\mathbf{z}_{t+1}|\mathbf{z}_t, \mathbf{a}_t) \prod_{t=1}^{\tau+1} p(\mathbf{x}_t|\mathbf{z}_t) \prod_{t=\tau+1}^{T} p(\mathcal{O}_t|\mathbf{z}_t, \mathbf{a}_t) \prod_{t=\tau+1}^{T} p(\mathbf{a}_t). \quad (13)$$

We use the posterior from Equation (12) and Jensen's inequality to obtain the ELBO of the marginal likelihood,

$$\log p(\mathbf{x}_{1:\tau+1}, \mathcal{O}_{\tau+1:T}|\mathbf{a}_{1:\tau})$$
$$= \log \int_{\mathbf{z}_{1:T}} \int_{\mathbf{a}_{\tau+1:T}} p(\mathbf{x}_{1:\tau+1}, \mathcal{O}_{\tau+1:T}, \mathbf{z}_{1:T}, \mathbf{a}_{\tau+1:T}|\mathbf{a}_{1:\tau}) \, \mathrm{d}\mathbf{z}_{1:T} \, \mathrm{d}\mathbf{a}_{\tau+1:T}$$
$$\geq \mathop{\mathbb{E}}_{(\mathbf{z}_{1:T}, \mathbf{a}_{\tau+1:T}) \sim q} \left[ \sum_{t=1}^{\tau+1} \log p(\mathbf{x}_t|\mathbf{z}_t) \right.$$
$$- \mathrm{D}_{\mathrm{KL}}\left(q(\mathbf{z}_1|\mathbf{x}_1) \parallel p(\mathbf{z}_1)\right) - \sum_{t=1}^{\tau} \mathrm{D}_{\mathrm{KL}}\left(q(\mathbf{z}_{t+1}|\mathbf{x}_{t+1}, \mathbf{z}_t, \mathbf{a}_t) \parallel p(\mathbf{z}_{t+1}|\mathbf{z}_t, \mathbf{a}_t)\right)$$
$$\left. + \sum_{t=\tau+1}^{T} \Big( r(\mathbf{z}_t, \mathbf{a}_t) + \log p(\mathbf{a}_t) - \log \pi(\mathbf{a}_t|\mathbf{z}_t) \Big) \right]. \quad (14)$$

Notice that the dynamics terms $\log p(\mathbf{z}_{t+1}|\mathbf{z}_t, \mathbf{a}_t)$ for $t \geq \tau + 1$ from the posterior and the prior cancel each other out in the ELBO.

The first part of the ELBO corresponds to the model objective. When using the parametric function approximators, the negative of it corresponds directly to the model loss in Equation (9).

The second part of the ELBO corresponds to the maximum entropy RL objective. We assume a uniform action prior, so the $\log p(\mathbf{a}_t)$ term is a constant term that can be omitted when optimizing

this objective. We use message passing to optimize this objective, with messages defined as

$$Q(\mathbf{z}_t, \mathbf{a}_t) = r(\mathbf{z}_t, \mathbf{a}_t) + \underset{\mathbf{z}_{t+1} \sim q(\cdot | \mathbf{x}_{t+1}, \mathbf{z}_t, \mathbf{a}_t)}{\mathbb{E}} [V(\mathbf{z}_{t+1})] \tag{15}$$

$$V(\mathbf{z}_t) = \log \int_{\mathbf{a}_t} \exp(Q(\mathbf{z}_t, \mathbf{a}_t)) \, \mathrm{d}\mathbf{a}_t. \tag{16}$$

Then, the maximum entropy RL objective can be expressed in terms of the messages as

$$\underset{(\mathbf{z}_{\tau+1:T}, \mathbf{a}_{\tau+1:T}) \sim q}{\mathbb{E}} \left[ \sum_{t=\tau+1}^{T} \left( r(\mathbf{z}_t, \mathbf{a}_t) - \log \pi(\mathbf{a}_t | \mathbf{z}_t) \right) \right]$$

$$= \underset{\mathbf{z}_{\tau+1} \sim q(\cdot | \mathbf{x}_{\tau+1}, \mathbf{z}_\tau, \mathbf{a}_\tau)}{\mathbb{E}} \left[ \underset{\mathbf{a}_{\tau+1} \sim \pi(\cdot | \mathbf{z}_{\tau+1})}{\mathbb{E}} \left[ Q(\mathbf{z}_{\tau+1}, \mathbf{a}_{\tau+1}) - \log \pi(\mathbf{a}_{\tau+1} | \mathbf{z}_{\tau+1}) \right] \right] \tag{17}$$

$$= \underset{\mathbf{z}_{\tau+1} \sim q(\cdot | \mathbf{x}_{\tau+1}, \mathbf{z}_\tau, \mathbf{a}_\tau)}{\mathbb{E}} \left[ -\mathrm{D}_{\mathrm{KL}} \left( \pi(\mathbf{a}_{\tau+1} | \mathbf{z}_{\tau+1}) \,\middle\|\, \frac{\exp\left(Q(\mathbf{z}_{\tau+1}, \mathbf{a}_{\tau+1})\right)}{\exp\left(V(\mathbf{z}_{\tau+1})\right)} \right) + V(\mathbf{z}_{\tau+1}) \right], \tag{18}$$

where the first equality is obtained from dynamic programming (see Levine (2018) for details), the second equality holds from the definition of KL divergence, and $\exp\left(V(\mathbf{z}_t)\right)$ is the normalization factor for $\exp\left(Q(\mathbf{z}_t, \mathbf{a}_t)\right)$ with respect to $\mathbf{a}_t$. Since the KL divergence term is minimized when its two arguments represent the same distribution, the optimal policy is given by

$$\pi(\mathbf{a}_t | \mathbf{z}_t) = \exp\left(Q(\mathbf{z}_t, \mathbf{a}_t) - V(\mathbf{z}_t)\right). \tag{19}$$

Noting that the KL divergence term is zero for the optimal action, the equality from Equation (18) can be used in Equation (15) to obtain

$$Q(\mathbf{z}_t, \mathbf{a}_t) = r(\mathbf{z}_t, \mathbf{a}_t) + \underset{\mathbf{z}_{t+1} \sim q(\cdot | \mathbf{x}_{t+1}, \mathbf{z}_t, \mathbf{a}_t)}{\mathbb{E}} \left[ \underset{\mathbf{a}_{\tau+1} \sim \pi(\cdot | \mathbf{z}_{\tau+1})}{\mathbb{E}} \left[ Q(\mathbf{z}_{t+1}, \mathbf{a}_{t+1}) - \log \pi(\mathbf{a}_{t+1} | \mathbf{z}_{t+1}) \right] \right]. \tag{20}$$

This equation corresponds to the standard Bellman backup with a soft maximization for the value function.

As mentioned in Section 5, our algorithm conditions the parametric policy in the history of observations and actions, which allows us to directly execute the policy without having to perform inference on the latent state at run time. When using the parametric function approximators, the negative of the maximum entropy RL objective, written as in Equation (17), corresponds to the policy loss in Equation (11). Lastly, the Bellman backup of Equation (20) corresponds to the Bellman residual in Equation (10) when approximated by a regression objective.

We showed that the SLAC objectives can be derived from applying variational inference in the control as inference framework in the POMDP setting. This leads to the joint likelihood of the past observations and future optimality variables, which we aim to optimize by maximizing the ELBO of the log-likelihood. We decompose the ELBO into the model objective and the maximum entropy RL objective. We express the latter in terms of messages of Q-functions, which in turn are learned by minimizing the Bellman residual. These objectives lead to the model, policy, and critic losses.

## B  NETWORK ARCHITECTURES

Recall that our full sequential latent variable model has two layers of latent variables, which we denote $\mathbf{z}_t^1$ and $\mathbf{z}_t^2$. We found this design to provide a good balance between ease of training and expressivity, producing good reconstructions and generations and, crucially, providing good representations for reinforcement learning. For reference, we reproduce the model diagram from the main paper in Figure 8. Note that this diagram represents the *Bayes net* corresponding to our full model. However, since all of the latent variables are stochastic, this visualization also

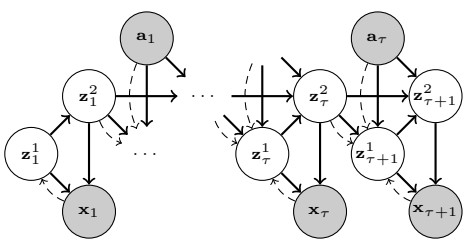

Figure 8: Diagram of our full model, reproduced from the main paper. Solid arrows show the generative model, dashed arrows show the inference model. Rewards are not shown for clarity.

presents the design of the computation graph. Inference over the latent variables is performed using amortized variational inference, with all training done via reparameterization. Hence, the computation graph can be deduced from the diagram by treating all solid arrows as part of the generative model and all dashed arrows as part of approximate posterior. The generative model consists of the following probability distributions, as described in the main paper:

$$
\begin{aligned}
\mathbf{z}_1^1 &\sim p(\mathbf{z}_1^1) \\
\mathbf{z}_1^2 &\sim p_\psi(\mathbf{z}_1^2|\mathbf{z}_1^1) \\
\mathbf{z}_{t+1}^1 &\sim p_\psi(\mathbf{z}_{t+1}^1|\mathbf{z}_t^2, \mathbf{a}_t) \\
\mathbf{z}_{t+1}^2 &\sim p_\psi(\mathbf{z}_{t+1}^2|\mathbf{z}_{t+1}^1, \mathbf{z}_t^2, \mathbf{a}_t) \\
\mathbf{x}_t &\sim p_\psi(\mathbf{x}_t|\mathbf{z}_t^1, \mathbf{z}_t^2) \\
r_t &\sim p_\psi(r_t|\mathbf{z}_t^1, \mathbf{z}_t^2, \mathbf{a}_t, \mathbf{z}_{t+1}^1, \mathbf{z}_{t+1}^2).
\end{aligned}
$$

The initial distribution $p(\mathbf{z}_1^1)$ is a multivariate standard normal distribution $\mathcal{N}(\mathbf{0}, \boldsymbol{I})$. All of the other distributions are conditional and parameterized by neural networks with parameters $\psi$. The networks for $p_\psi(\mathbf{z}_1^2|\mathbf{z}_1^1)$, $p_\psi(\mathbf{z}_{t+1}^1|\mathbf{z}_t^2, \mathbf{a}_t)$, $p_\psi(\mathbf{z}_{t+1}^2|\mathbf{z}_{t+1}^1, \mathbf{z}_t^2, \mathbf{a}_t)$, and $p_\psi(r_t|\mathbf{z}_t^1, \mathbf{z}_t^2, \mathbf{a}_t, \mathbf{z}_{t+1}^1, \mathbf{z}_{t+1}^2)$ consist of two fully connected layers, each with 256 hidden units, and a Gaussian output layer. The Gaussian layer is defined such that it outputs a multivariate normal distribution with diagonal variance, where the mean is the output of a linear layer and the diagonal standard deviation is the output of a fully connected layer with softplus non-linearity. The observation model $p_\psi(\mathbf{x}_t|\mathbf{z}_t^1, \mathbf{z}_t^2)$ consists of 5 transposed convolutional layers (256 $4 \times 4$, 128 $3 \times 3$, 64 $3 \times 3$, 32 $3 \times 3$, and 3 $5 \times 5$ filters, respectively, stride 2 each, except for the first layer). The output variance for each image pixel is fixed to 0.1.

The variational distribution $q$, also referred to as the inference model or the posterior, is represented by the following factorization:

$$
\begin{aligned}
\mathbf{z}_1^1 &\sim q_\psi(\mathbf{z}_1^1|\mathbf{x}_1) \\
\mathbf{z}_1^2 &\sim p_\psi(\mathbf{z}_1^2|\mathbf{z}_1^1) \\
\mathbf{z}_{t+1}^1 &\sim q_\psi(\mathbf{z}_{t+1}^1|\mathbf{x}_{t+1}, \mathbf{z}_t^2, \mathbf{a}_t) \\
\mathbf{z}_{t+1}^2 &\sim p_\psi(\mathbf{z}_{t+1}^2|\mathbf{z}_{t+1}^1, \mathbf{z}_t^2, \mathbf{a}_t).
\end{aligned}
$$

Note that the variational distribution over $\mathbf{z}_1^2$ and $\mathbf{z}_{t+1}^2$ is intentionally chosen to exactly match the generative model $p$, such that this term does not appear in the KL-divergence within the ELBO, and a separate variational distribution is only learned over $\mathbf{z}_1^1$ and $\mathbf{z}_{t+1}^1$. This intentional design decision simplifies the inference process. The networks representing the distributions $q_\psi(\mathbf{z}_1^1|\mathbf{x}_1)$ and $q_\psi(\mathbf{z}_{t+1}^1|\mathbf{x}_{t+1}, \mathbf{z}_t^2, \mathbf{a}_t)$ both consist of 5 convolutional layers (32 $5 \times 5$, 64 $3 \times 3$, 128 $3 \times 3$, 256 $3 \times 3$, and 256 $4 \times 4$ filters, respectively, stride 2 each, except for the last layer), 2 fully connected layers (256 units each), and a Gaussian output layer. The parameters of the convolution layers are shared among both distributions.

The latent variables have 32 and 256 dimensions, respectively, i.e. $\mathbf{z}_t^1 \in \mathbb{R}^{32}$ and $\mathbf{z}_t^2 \in \mathbb{R}^{256}$. For the image observations, $\mathbf{x}_t \in [0, 1]^{64 \times 64 \times 3}$. All the layers, except for the output layers, use leaky ReLU non-linearities. Note that there are no deterministic recurrent connections in the network—all networks are feedforward, and the temporal dependencies all flow through the stochastic units $\mathbf{z}_t^1$ and $\mathbf{z}_t^2$

For the reinforcement learning process, we use a critic network $Q_\theta$ consisting of 2 fully connected layers (256 units each) and a linear output layer. The actor network $\pi_\phi$ consists of 5 convolutional layers, 2 fully connected layers (256 units each), a Gaussian layer, and a tanh bijector, which constrains the actions to be in the bounded action space of $[-1, 1]$. The convolutional layers are the same as the ones from the latent variable model, but the parameters of these layers are not updated by the actor objective. The same exact network architecture is used for every one of the experiments in the paper.

| Benchmark | Task | Action repeat | Original control time step | Effective control time step |
|---|---|---|---|---|
| DeepMind Control Suite | cheetah run | 4 | 0.01 | 0.04 |
| | walker walk | 2 | 0.025 | 0.05 |
| | ball-in-cup catch | 4 | 0.02 | 0.08 |
| | finger spin | 2 | 0.02 | 0.04 |
| OpenAI Gym | HalfCheetah-v2 | 1 | 0.05 | 0.05 |
| | Walker2d-v2 | 4 | 0.008 | 0.032 |
| | Hopper-v2 | 2 | 0.008 | 0.016 |
| | Ant-v2 | 4 | 0.05 | 0.2 |

Table 1: Action repeats and the corresponding agent's control time step used in our experiments.

## C  TRAINING AND EVALUATION DETAILS

The control portion of our algorithm uses the same hyperparameters as SAC (Haarnoja et al., 2018a), except for a smaller replay buffer size of 100000 environment steps (instead of a million) due to the high memory usage of image observations. All of the parameters are trained with the Adam optimizer (Kingma & Ba, 2015), and we perform one gradient step per environment step. The Q-function and policy parameters are trained with a learning rate of 0.0003 and a batch size of 256. The model parameters are trained with a learning rate of 0.0001 and a batch size of 32. We use sequences of length $\tau = 8$ for all the tasks. Note that the sequence length can be less than $\tau$ for the first $t$ steps ($t < \tau$) of each episode.

We use action repeats for all the methods, except for D4PG for which we use the reported results from prior work (Tassa et al., 2018). The number of environment steps reported in our plots correspond to the unmodified steps of the benchmarks. Note that the methods that use action repeats only use a fraction of the environment steps reported in our plots. For example, 3 million environment steps of the cheetah task correspond to 750000 samples when using an action repeat of 4. The action repeats used in our experiments are given in Table 1.

Unlike in prior work (Haarnoja et al., 2018a;b), we use the same stochastic policy as both the behavioral and evaluation policy since we found the deterministic greedy policy to be comparable or worse than the stochastic policy.

## D  ADDITIONAL PREDICTIONS FROM THE LATENT VARIABLE MODEL

We show additional samples from our model in Figure 9 and Figure 10. Samples from the posterior show the images $\mathbf{x}_t$ as constructed by the decoder $p_\psi(\mathbf{x}_t|\mathbf{z}_t)$, using a sequence of latents $\mathbf{z}_t$ that are encoded and sampled from the posteriors, $q_\psi(\mathbf{z}_1|\mathbf{x}_1)$ and $q_\psi(\mathbf{z}_{t+1}|\mathbf{x}_{t+1},\mathbf{z}_t,\mathbf{a}_t)$. Samples from the prior, on the other hand, use a sequence of latents where $\mathbf{z}_1$ is sampled from $p(\mathbf{z}_1)$ and all remaining latents $\mathbf{z}_t$ are from the propagation of the previous latent state through the latent dynamics $p_\psi(\mathbf{z}_{t+1}|\mathbf{z}_t,\mathbf{a}_t)$. These samples do not use any image frames as inputs, and thus they do not correspond to any ground truth sequence. We also show samples from the conditional prior, which is conditioned on the first image from the true sequence: for this, the sampling procedure is the same as the prior, except that $\mathbf{z}_1$ is encoded and sampled from the posterior $q_\psi(\mathbf{z}_1|\mathbf{x}_1)$, rather than being sampled from $p(\mathbf{z}_1)$.

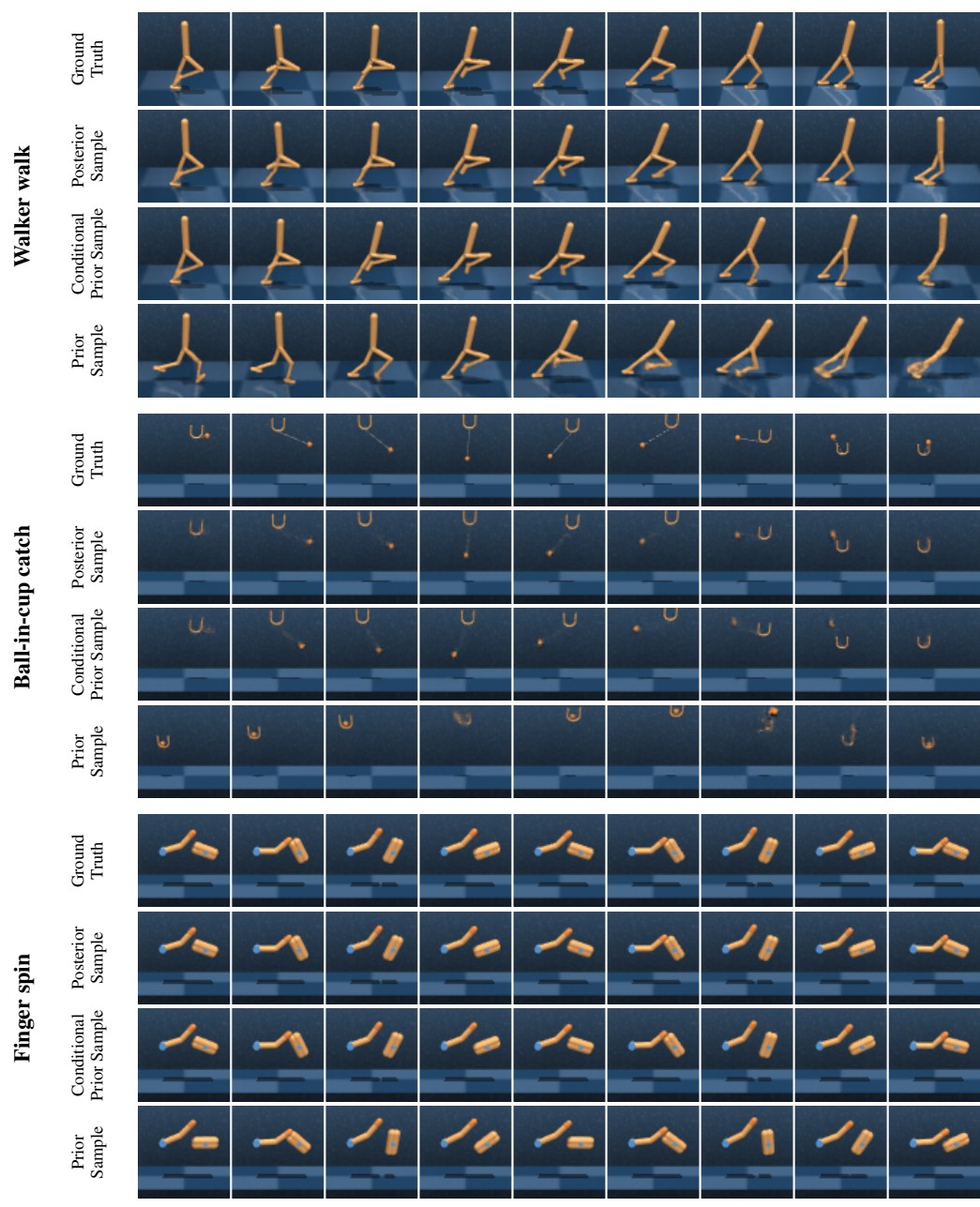

Figure 9: Example image sequences, along with generated image samples, for three of the DM Control tasks that we used in our experiments. See Figure 7 for more details and for image samples from the cheetah task.

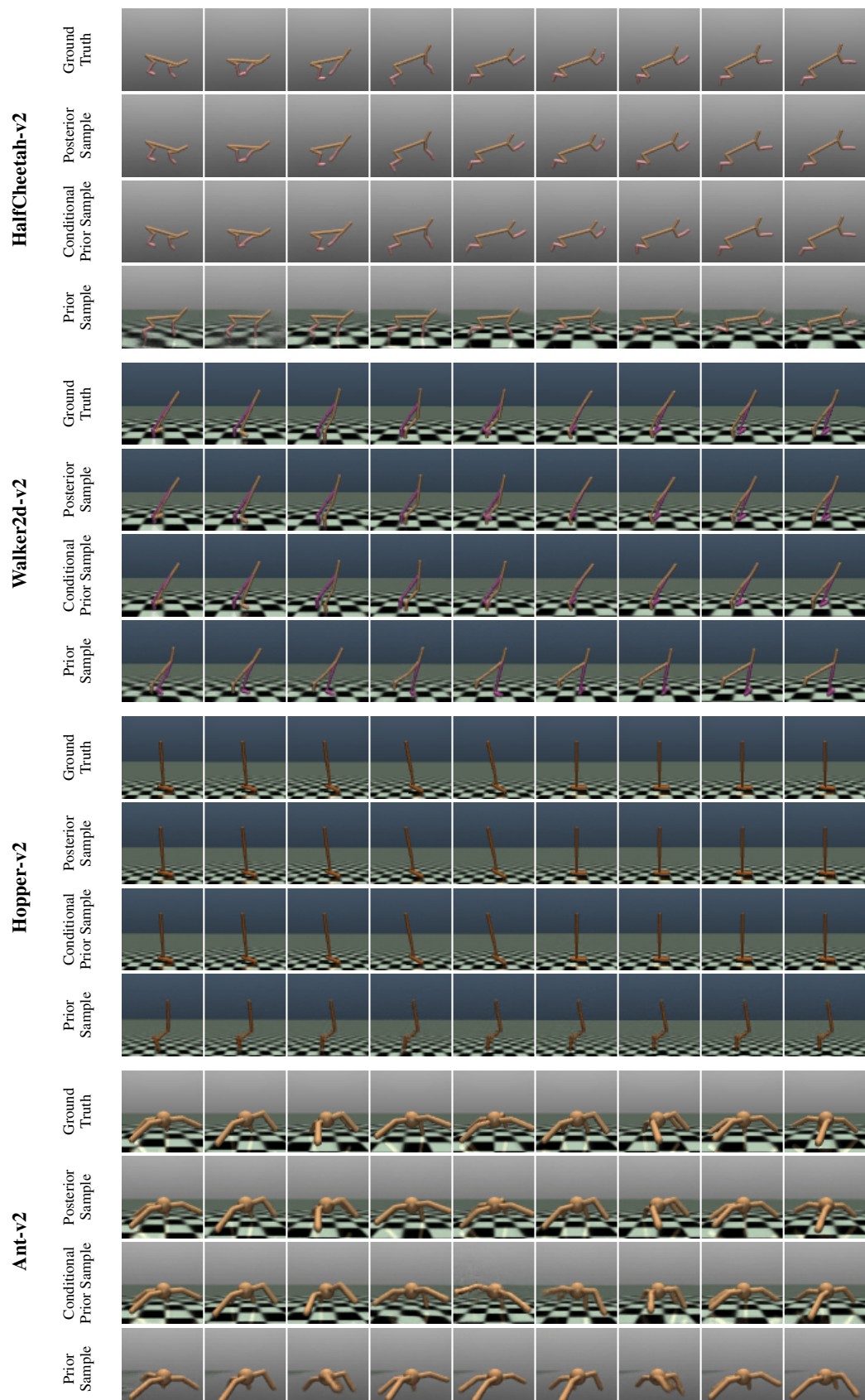

Figure 10: Example image sequences, along with generated image samples, for the four OpenAI Gym tasks that we used in our experiments.

