# OpenReview forum: "Stochastic Latent Actor-Critic: Deep Reinforcement Learning with a Latent Variable Model"
_ICLR.cc/2020/Conference — Reject_

### Official Review · AnonReviewer3 · 2019-10-22
**Official Blind Review #3**

**Rating:** 3

**Review:**

This work proposed a fully stochastic RL method and demonstrated significantly improved performance on multiple tasks.

Pros:
1. The presentation is very clear and easy to read.
2. The proposed method is sensible
3, The experimental evaluation shows great practical gain

Cons:
1. The method itself is incremental. As discussed in the related work, this work can be seen as complementary to many related works such as Igl 18, but the novelty of the idea is rather limited.
2. The claims and the real-benefit of the method may not be consistent. (My biggest concern)
The paper claims that full stochasticity contributed to the practical gain but in the experiment Figure 6, we can see the simple filtering does not perform well.
It seems that the benefit of the method is rather from such particular latent space design rather than the stochastic vs deterministic.
3. Continue with the previous point, Figure 2 is not very well motivated and I believe that from Figure 1 to figure 2 design was the most important part of the performance gain. Such important designed was very briefly described without any motivation.
4. With the previous point, the experiments may be unfair, because, another partially stochastic method can easily utilize such design and further improve the performance.
5. The related work should add a discussion about stochastic sequential models such as Kalman VAE etc.  paragraph 3 motivates your contribution as VAE does not model sequential information. But there are many works such as the KVAE that are stochastic and models sequential information.



**Experience Assessment:**

I have read many papers in this area.

**Review Assessment: Checking Correctness Of Derivations And Theory:**

I assessed the sensibility of the derivations and theory.

**Review Assessment: Checking Correctness Of Experiments:**

I assessed the sensibility of the experiments.

**Review Assessment: Thoroughness In Paper Reading:**

I made a quick assessment of this paper.

---

> ### Author Response · Authors · 2019-11-14
> **Author Reply for Official Blind Review #3**
>
> We thank the reviewer for the constructive feedback. In this response, we clarify the novelty of our method and the distinction with prior work, and we emphasize that the derivation of our method is novel and sound (we edited the paper to make this more clear). We also revised our paper to include additional references to prior work. Please see the answers below for clarifications.
>
> - "method itself is incremental"
> To the best of our knowledge, our work is the first model-free RL method for POMDPs that shows that the critic can directly be conditioned on individual latent states sampled from a stochastic model. This realization is non-trivial and novel, and our paper provides justification for it (we revised the paper to include a more detailed derivation in Appendix A of the revised paper). In contrast, prior model-free RL approaches convert the POMDP into an MDP by redefining the state space, and then performing RL in the converted MDP (e.g. in the space of learned belief representations or the history of observations and actions). An interesting finding of our derivation is that we can just sample z_t and z_{t+1} from the posterior and use those samples for the backup, instead of performing probabilistic filtering of the latent belief.
>
> - "benefit of the method is rather from such particular latent space design rather than the stochastic vs deterministic".
> The deterministic model has the same latent space factorization as our model, thus controlling for the latent space design. We revised the text in Section 7.2 to clarify this. The results from Figure 6 indicate that although the particular factorization provided benefits (fully/partially stochastic outperforms simple filtering), the stochasticity of the model also contributed to the benefits (fully/partially stochastic outperforms deterministic).
>
> - "this work can be seen as complementary to many related works such as Igl 18".
> As noted above regarding novelty, prior works perform RL on a belief representation, whereas our work shows that it is possible to train a critic directly on latent states. In the case of Igl 18, they use particle filtering to propagate the belief forward, and then encode the particles into a belief representation, which is then used for the actor and the critic. In addition, we focus on tasks with high-dimensional image observations for complex underlying continuous control tasks, in contrast to Igl 18, which evaluates on Atari tasks and low-dimensional continuous control tasks that emphasize partial observability and knowledge-gathering actions.
>
> - Related works.
> We updated Section 2 to include additional references of prior work that studies stochastic sequential models.
>
> - "the experiments may be unfair, because, another partially stochastic method can easily utilize such design and further improve the performance".
> We believe the comparison is fair since each algorithm uses a model that was chosen to work well with its particular algorithm, e.g. the PlaNet model was likely chosen for the quality of its future reward predictions, whereas our model was chosen for the quality of its representations.
>
> - Motivation of factorization of our latent variable.
> We draw motivation from recent success of autoregressive latent variables in VAEs (VQ-VAE2, Razavi et al. 2019; BIVA, Maaloe et al. 2019). This factorization results in latent distributions that are more expressive, and it allows for some parts of the prior and posterior distributions to be shared. We added this at the beginning of Section 6 of the revised paper.

---

### Official Review · AnonReviewer2 · 2019-10-24
**Official Blind Review #2**

**Rating:** 8

**Review:**

The authors propose SLAC, an important extension of the recently introduced soft actor critic (SAC) algorthm, which operates on a learned latent state, rather than an observed one, and therefore aims to jointly learn to represent high dimensional inputs and execute continuous control based on this representation.

SLAC is based on a dynamic, non-linear Markov generative model and incorporates structured variational inference to learn the latent state distribution. The generative model and amortized variational inference support the learning of strong latent expected future reward estimates (Q functions that condition on the latent state), which the policy, which conditions directly on the observations (i.e. image) is distilled against for fast inference. The paper demonstrates solid gains over existing techniques, brings together recent work under a rigorous framework, and is a pleasure to read.

Strengths:
-Novel formulation, SOTA results, well written.

Limitations:
-While the most important ablation, the role of making the primary latent variable stochastic, is investigated, a deeper investigation of what makes the model more effective than existing techniques would be insightful, and further strengthen the paper.
-Related, the approach seems closest to PlaNet in structure, but rather than being used for planning, is executed directly as an off-policy actor-critic algorithm, generalizing SAC. A discussion, and possibly some additional experiments to explain the differences and understand the tradeoffs would strengthen the paper. The authors mention ", contrary to the conclusions in prior work (Hafner et al., 2019; Buesing et al., 2018), the fully stochastic model performs on par or better." Why?

Minor:
-Figure 6 partially stochastic in figure, mixed in text.

Overall:
A strong paper, that brings together and generalizes existing work, with strong experimentation and SOTA results. Definite accept.


**Experience Assessment:**

I have published one or two papers in this area.

**Review Assessment: Checking Correctness Of Derivations And Theory:**

I assessed the sensibility of the derivations and theory.

**Review Assessment: Checking Correctness Of Experiments:**

I assessed the sensibility of the experiments.

**Review Assessment: Thoroughness In Paper Reading:**

I read the paper at least twice and used my best judgement in assessing the paper.

---

> ### Author Response · Authors · 2019-11-14
> **Author Reply for Official Blind Review #2**
>
> We thank the reviewer for the comments and feedback. We have revised the paper to address the points below.
>
> - '"contrary to the conclusions in prior work (Hafner et al., 2019; Buesing et al., 2018), the fully stochastic model performs on par or better." Why?'
> We revised the results from Figure 6 and the text in Section 7.2 to adjust the claims about the importance of various design decisions (the fully and partially stochastic models both perform equally well). We hypothesize that these prior works benefit from the deterministic paths (realized as an LSTM or GRU) because they use multi-step samples from the prior. In contrast, our method uses samples from the posterior, which are conditioned on same-step observations, and thus they are less sensitive to the propagation of the latent states through time. We revised the text in Section 7.2. to include this reasoning.
>
> - "explain the differences and understand the tradeoffs" between PlaNet and SLAC.
> The similarity between PlaNet (and other model-based methods) and SLAC is that they both learn a latent variable model, while the main difference among them is whether the method is model-based or model-free. We revised the second paragraph of Section 2 to expand on the differences and tradeoffs between this type of methods and ours.
>
> - "Figure 6 partially stochastic in figure, mixed in text."
> We revised the paper to consistently refer to this variant as "partially stochastic".

---

### Official Review · AnonReviewer4 · 2019-11-02
**Official Blind Review #4**

**Rating:** 3

**Review:**

This paper proposes an actor-critic method that tries to aid learning good policies via learning a good representation of the state space (via a latent variable model). In actor-critic methods, the critic is learnt to evaluate policies in the latent space, which further helps with efficient policy optimization. The proposed method is evaluated on image-based control tasks, with baseline evaluations against both model-based and model-free methods in terms of sample efficiency.

- The key argument is that learning policies in the latent space is more efficient, as it is possible to learn good representations in the latent space. There are quite a few recent works (e.g DeepMDP, Gelada et al., 2019;  Dadashi et al., 2019) that talks about representation learning in RL, and yet the paper makes no relations or references to previous works. I find it surprising that none of the past related works are mentioned in the paper.

- I find the arguments on solving a POMDP instead of a MDP a bit vague in this context. I understand that the goal is to solve image based control tasks - for which learning good representations via a latent variable model might be useful, but it does not explicitly require references to a POMDP? In most ALE tasks, we have pixel based observations too, which makes the ALE environments a POMDP in some sense, but we use approximations to it to make it equivalent to a MDP with sufficient history. The arguments on POMDP seems rather an additional mention, with no necessary significance to it?

- The paper mentions solving RL in the learned latent space, which is empirically proposed to be a good approach without theoretical justifications. There are several recent works that tries to understand the representation learning in RL problem from a theoretical perspective too - it would be useful to see where this approach stands in light of those theoretical results? Otherwise, the contribution seems rather limited : solving RL in latent space is useful, but there are no justifications to it? Why should this approach even be adapted or what is the significance of it?

- The proposed actor-critic method in the latent space is built on top of Soft Actor-Critic (SAC). I understand this is a design/implementation approach building from previous works - but it would have been useful to add more context as to what it means to learn a critic in the latent space. If the critic evaluates a policy in the latent space - then is this a good policy evaluation for actor-critic itself? Why or why not? I do not understand why the critic evaluation in the latent space is even a good approach?

- My first impression was that the paper proposes a separate auxilliary objective for learning good representations based on which actor-critic algorithms can be made more efficient. However, this does not seem to be the case directly? Following on previous point - I find the argument of solving a critic in the latent space rather vague.

- The sequential latent variable model proposed is based on existing literature. This can be any latent variable model (e.g VAEs), but I understand, as mentioned in the paper, the design choice of using sequential models to capture the temporal aspect.

- The proposed algorithm is in fact a combination of SAC and sequential latent variable models, both of which are well-known in the literature. The SLAC algorithm combines these to solve image-based control tasks. As per equation 10, which is the regular policy optimization objective with max entropy - the only difference is that the critic is evaluated in the latent space. This appears to me as more of an engineeering choice, and experimentally one that perhaps give good results - but the lack of justifications of why equation 10 is even the right objective to solve makes the paper rather less appealing.

- I think overall the contribution of the paper is rather limited. It is more of an experimental design and engineering approach that combines previous known techniques. The paper mentions learning good representations for RL, without any references or justifications - and it appears that overall there are bold claims made in the paper but it lacks significant scientific contribution.

- Experimental evaluations are made on image based control tasks. Experimental results are compared to few baselines - but it is not clear whether these are even the right baselines. For example, it would have been good to include analysis of the proposed model with different latent variable models (including VAE) to perhaps justify the choice of the latent variable model. Results in figure 5 appear a bit concerning to me - these are mostly the standard Mujoco tasks from the OpenAI suite. Are these all image based benchmarks too, or the standard baselines? It is not clear from the text. Assuming they are standard baselines, the comparisons made are rather unfair (for example : SAC and MDP performs much better on tasks like HalfCheetah-v2). Why are the baselines performing so poorly in the results?

- Overall, I think the paper needs more work in terms of writing and justifying the choice of the approach. There are significant references missing in the paper. Most importantly, there are quite a few claims made in the paper which are not properly justified, that makes the overall contribution and novelty of the paper rather limited. I would tend for a rejection of this paper, as it requires more work - both in terms of theoretical justifications (including references) and experimetnal ablation studies and more simpler benchmarks explaining the choice of the approach.



**Experience Assessment:**

I have published one or two papers in this area.

**Review Assessment: Checking Correctness Of Derivations And Theory:**

I assessed the sensibility of the derivations and theory.

**Review Assessment: Checking Correctness Of Experiments:**

I carefully checked the experiments.

**Review Assessment: Thoroughness In Paper Reading:**

I read the paper thoroughly.

---

> ### Author Response · Authors · 2019-11-14
> **Author Reply for Official Blind Review #4**
>
> We thank the reviewer for the detailed comments and feedback. In this response, we clarify the novelty of our method and the distinction with prior work, and we emphasize that the derivation of our method is novel and sound (we edited the paper to make this more clear). We also revised our paper to include additional references to prior work. Please see the answers below for clarifications.
>
> - Novelty.
> To the best of our knowledge, our work is the first model-free RL method for POMDPs that shows that the critic can directly be conditioned on individual latent states sampled from a stochastic model. This realization is non-trivial and novel, and our paper provides justification for it (see next paragraph). In contrast, prior model-free RL approaches convert the POMDP into an MDP by redefining the state space, and then performing RL in the converted MDP (e.g. in the space of learned belief representations or the history of observations and actions). An interesting finding of our derivation is that we can just sample z_t and z_{t+1} from the posterior and use those samples for the backup, instead of performing probabilistic filtering of the latent belief.
>
> - Learning the critic in the latent space "without theoretical justifications", "lack of justifications of why equation 10 is even the right objective".
> We revised the paper to emphasize that this choice follows from the ELBO (Section 5, first paragraph) and to include a more detailed derivation of the ELBO and the Bellman backup for POMDPs (Appendix A in the revised paper). The choice of learning the critic in the latent space follows from approximately maximizing the ELBO of the log-likelihood of the past observations and future optimality variables. The model loss (Eq. 9) corresponds to the first part of the ELBO and the policy loss (Eq. 11) corresponds to the second part of the ELBO. The Bellman residual (Eq. 10) is an approximation to the Bellman backups of the Q-function (Appendix A in the revised paper provides justification for the latent-space Bellman backup).
>
> - Related works.
> We updated Section 2 to include additional references of prior work that studies representation learning in RL.
>
> - "include analysis of the proposed model with different latent variable models (including VAE)".
> We revised the results from Figure 6, in which we compare our algorithm with various versions of the model (including VAE), and we revised the text in Section 7.2 to adjust the claims about the importance of various design decisions accordingly. We found that our fully stochastic model and the partially stochastic model both perform equally well. The performance was worse when using a model without the 2-level factorization, and significantly worse when using a standard per-time-step VAE with no temporal dependencies. The primary purpose of this ablation experiment is to analyze which components of SLAC’s latent variable model were important for its performance.
>
> - "Are these all image based benchmarks too" (for OpenAI gym tasks).
> All of these results are on image-based benchmarks unless it's labeled as "state" (the only one being "SAC (state)" in Figure 4). We revised Figure 4 and 5 with longer learning curves and with captions that specify that the experiments are from images (unless otherwise noted in Figure 4). We also now include the asymptotic performance of MPO, which is competitive to our approach for some tasks, though the sample efficiency is not.
>
> - "Why are the baselines performing so poorly in the results?" (for OpenAI gym tasks).
> Prior papers do not report results for off-policy model-free RL algorithms from images on MuJoCo gym tasks, and very few prior works report image-based results on DeepMind control continuous tasks. We made our best effort to choose good hyperparameters for prior methods on these tasks, and we believe that these results reflect their actual performance. Image-based control on these tasks is quite difficult.

---

### Decision · Program_Chairs · 2019-12-19

**Decision:**

Reject

**Comment:**

An actor-critic method is introduced that explicitly aims to learn a good representation using a stochastic latent variable model. There is disagreement among the reviewers regarding the significance of this paper. Two of the three reviewers argue that several strong claims made in the paper that are not properly backed up by evidence. In particular, it is not sufficiently clear to what degree the shown performance improvement is due to the stochastic nature of the model used, one of the key points of the paper. I recommend that the authors provide more empirical evidence to back up their claims and then resubmit.